# ON THE IMPACT OF HYPER-PARAMETERS ON THE PRIVACY OF DEEP NEURAL NETWORKS

## ABSTRACT

The deployment of deep neural networks (DNNs) in many real-world applications leads to the processing of huge amounts of potentially sensitive data. This raises important new concerns, in particular with regards to the privacy of individuals whose data is used by these DNNs. In this work, we focus on DNNs trained to identify biometric markers from images, e.g., gender classification, which have been shown to leak unrelated private attributes at inference time, e.g., ethnicity, also referred to as unintentional feature leakage. Specifically, we observe that the hyper-parameters of such DNNs significantly impact the leakage of these attributes unrelated to the main task. To address this, we develop a hyper-parameter optimization (HPO) strategy with the goal of training DNNs that mitigate unintended feature leakage while retaining a good main task accuracy. We use a multi-fidelity and multi-objective HPO approach to (i) conduct the first study of the impact of hyper-parameters on the risk of unintended feature leakage; (ii) demonstrate that, for a specific main task, HPO successfully identifies hyper-parameter configurations that considerably reduce the privacy risk at a very low impact on utility; and (iii) evidence that there exist hyper-parameter configurations that have a significant impact on the privacy risk, regardless of the choice of main and private tasks, i.e., hyper-parameters that generally better preserve privacy.

## 1 INTRODUCTION

Deep neural networks (DNNs) have led to tremendous strides in visual recognition, addressing diverse tasks such as image classification (He et al., 2016), object detection (Ren et al., 2015), semantic segmentation (Chen et al., 2018), human pose estimation (Kanazawa et al., 2018), and 3D reconstruction (Dou et al., 2017). As such, they have now reached the point of being deployed in many real-world applications, such as automated driving, surveillance and security, and image content analysis in social media.

Many of these models are deployed on cloud servers and offer their functionality as a service for users to send their data to and get a prediction in return. This, however, raises privacy concerns, as images often contain personal information about these users, which is not related to the main task they are querying, and that they would like to keep private. A widely-used approach to limit the information unrelated to the main task contained in the user input is to instead let the user send embeddings, i.e., features extracted by a local DNN, to the server. This, for example, is the solution provided by services such as Azure AI Services or Amazon's AWS. Concrete examples of this scenario include mobile applications offering services based on data collection on the user side and analysis on the server side, such as location-based services (Waze, 2025; UberEats, 2025), and healthcare and biometrics application (Flo, 2025; Aware, 2025). While safer than sending images directly to the cloud, it has been shown that the features extracted by a DNN trained for a specific task contain auxiliary information the user may not have consented to share with the service provider (Melis et al., 2018; Özbulak et al., 2016; Das et al., 2018; Parde et al., 2019; Terhörst et al., 2020). Having access to these embeddings, an attacker, or even the service provider itself, may be able to infer private user attributes, as depicted in Fig. 1.

In this paper, we therefore place ourselves in the adversarial scenario where an attacker is able to intercept the features extracted by a DNN trained for a specific task, and ask ourselves: Can one construct and train the DNN such that its features leak the least amount of private information pos-

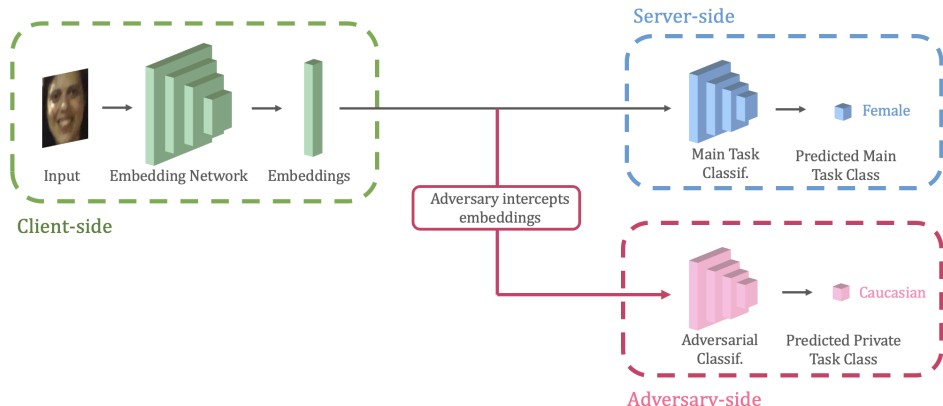

Figure 1: Example of unintended feature leakage in a cloud service for gender classification. Under our described adversarial model, the adversary is able to predict the private attribute "Ethnicity" when accessing to the embeddings output by the Embedding Network on the client side.

sible while preserving what is necessary for the main task and therefore limit the risk of unintended feature leakage?

This adversarial scenario, which focuses on the privacy leakage of embeddings at inference time, is common in the privacy-preserving representation learning field (Li et al., 2019; Stadler et al., 2024; Osia et al., 2017; 2018). Previous works have tackled this problem via adversarial and disentangled representation learning (Bertran et al., 2019; Li et al., 2021; Wang et al., 2024; Morales et al., 2021; Bortolato et al., 2020). However, the resulting methods are usually highly task-specific and require a specific architecture or training process, which complicates their generalization to other tasks. Indeed, deploying these techniques in cloud services will not be seamless, deterring service providers from putting them into practice, especially if privacy is not a priority.

By contrast, we focus on a highly generalizable aspect of DNNs, which is architecture-agnostic: The hyper-parameters used to train DNNs. Hyper-parameters have been shown countless times to have a significant impact on the accuracy of the main task (Wong et al., 2019; Bergstra et al., 2013; Hutter et al., 2013; 2014; Liao et al., 2022), which has lead to the development and surge of hyper-parameter-optimization (HPO) methods (O'Malley et al., 2019; Falkner et al., 2018), aiming to find the configuration of hyper-parameters leading to the best performing models in terms of accuracy. Additionally, hyper-parameters have also been shown to have an impact on other metrics such as fairness or energy consumption (Sukthanker et al., 2023a). In this paper, we study the impact of hyper-parameters on **privacy** and leverage multi-objective HPO (Knowles, 2006) to find Pareto-dominant models in terms of main task accuracy and privacy.

We therefore aim to place privacy at the same level of importance as utility. This paradigm, also referred to as privacy-by-design, is crucial in privacy engineering, and encourages putting privacy to the forefront when designing any system. We hope therefore to better guide the ML community and the industry towards more privacy-preserving models with the following **contributions**:

- Focusing on DNNs whose main task is a biometric classification task, e.g., gender classification, we conduct the first study of the impact of hyper-parameters on the risk of unintended feature leakage (privacy risk).
- We introduce the first multi-objective HPO strategy to both maximize the main task accuracy and minimize the risk of unintended feature leakage, and show that it is possible to train models that better preserve privacy at a negligible utility cost.
- We observe that the hyper-parameters that have the most impact on the privacy risk for a specific private task also have a similar impact on other choices of main and private tasks.

We will make our code and models publicly available upon acceptance.

## 2 RELATED WORK

### 2.1 PRIVACY-PRESERVING MACHINE LEARNING

The vast majority of the attempts at introducing privacy into machine learning (ML) can be grouped into three categories: The methods that alter the input sample, e.g., with noise (Terhörst et al., 2019), generative adversarial networks (Mirjalili et al., 2020; Wang et al., 2021), or adversarial perturbations (Chhabra et al., 2018; Shan et al., 2020; Cilloni et al., 2022); those that follow an adversarial (Bertran et al., 2019; Roy & Boddeti, 2019; Li et al., 2021; Morales et al., 2021) or disentangled (Bortolato et al., 2020; Wang et al., 2024) representation learning strategy, aiming to retain information about the task of interest while discarding that about the private soft-biometrics; and finally methods relying on differential privacy, via perturbation either of the input or output (Fukuchi et al., 2017; Lowy & Razaviyayn, 2024), or of the training algorithm (Birrell et al., 2024).

While effective, these techniques suffer from drawbacks, such the fact that they can be detected (Rot et al., 2022) or even attacked (Osorio-Roig et al., 2021), but more importantly the trade-off between main task accuracy and privacy, as most of these methods degrade utility significantly. Additionally, most of the existing methods are very task specific or hard to deploy in practice, for example because they use embedding networks with very specific architectures (Bortolato et al., 2020; Wang et al., 2024), or complex adversarial training methods that rely on hard to tune hyper-parameters (Sukthanker et al., 2023a). By contrast, we focus on developing privacy-preserving training strategies that do not require modifying the desired architecture, or change the training strategy beyond hyper-parameters. While finding the optimal hyper-parameters for a given architecture through HPO is costly, our large-scale empirical study on the impact of hyper-parameters on privacy allows us to define general rules on the most privacy-preserving configurations, thus considerably facilitating the deployment process.

Some previous works have studied the impact of hyper-parameters in the context of privacy, but focus on very different privacy tasks. For example, Arous et al. (2023) and Tan et al. (2022) study the impact of hyper-parameters and over-parameterization on membership inference attacks (MIA), which aim to infer if a target data sample was used to train the model or not. Hyper-parameters have been shown to be directly linked to overfitting, which in turn has been proven to directly impact the success of MIA attacks and therefore the privacy of members of the training set (van Breugel et al., 2023). Although motivated by these results, we explore the impact of hyper-parameters on entirely different types of privacy risks, namely **unintended feature leakage**. This is a very different setup, as we do not focus on privacy w.r.t. the training set, but on the privacy risk at **inference time**. In contrast with the rather intuitive results for MIA, the link between training hyper-parameters and the risk of unintended feature leakage at inference time remains largely unexplored.

Developing privacy-preserving technology comes with an inherent privacy-utility trade-off, which is impossible to avoid. This is demonstrated in (Stadler et al., 2024), which claims that "it is not possible to learn representations that have high utility for the intended task but, at the same time, prevent inference of any attribute other than the task label itself". Regardless, we propose a method that identifies the training hyper-parameters of DNNs that will decrease the privacy risk *as much as possible* while maintaining a very similar main-task accuracy. Additionally, we make the privacy-utility trade-off observable, therefore enabling more educated decisions about how much privacy is considered when developing models.

### 2.2 HYPER-PARAMETER OPTIMIZATION (HPO)

HPO refers to the field aiming to automate the search for the optimal hyper-parameters, including batch size, learning rate, optimizer, loss function, and in some cases architectural choices. Work on HPO has mostly focused on finding training strategies to maximize performance (Liu et al., 2019; Nekrasov et al., 2019). Some methods have nonetheless been developed for multi-objective HPO, to also optimize for a secondary objective, such as power consumption, model size, latency, or even fairness (Tian et al., 2021). In particular, (Sukthanker et al., 2023b) jointly uses HPO and neural architecture search to optimize for both performance and fairness. However, to the best of our knowledge, such studies have never been conducted with privacy as a goal.

## 3 METHOD

### 3.1 MULTI-OBJECTIVE AND MULTI-FIDELITY HPO

Our goal is to find models that mitigate unintended feature leakage while preserving privacy by only changing their training hyper-parameters. This will enable us to study the impact of these hyper-parameters on the privacy risk of the features produced by an embedding network trained for a specific task. To do so we leverage HPO techniques to find hyper-parameter configurations that have the most impact on privacy while preserving utility.

Traditional HPO focuses on finding the hyper-parameter configurations that lead to the models with the best main task accuracy. Here, however, we seek to optimize for *both* accuracy and privacy. Doing so requires the following two properties: **(i) Multi-objective**: in contrast to classical HPO that optimizes for main task performance only, the goal is to optimize for two or more objectives, this would enable the observation of the trade-off between accuracy and privacy; **(ii) multi-fidelity**: evaluating the accuracy and privacy of a hyper-parameter configuration requires fully training a deep neural network. This can be computationally expensive, which motivates the need to approximate the conventional evaluation methods.

To achieve our goals, we use the SMAC3 package (Lindauer et al., 2021) that offers a framework satisfying both properties using a HyperBand-based algorithm (Li et al., 2016) called Sequential Surrogate Model-Based Optimization (Audet et al., 2000) for multi-fidelity and using the ParEGO algorithm (Knowles, 2006) for multi-objective.

### 3.2 METRICS

Before exploring the impact of hyper-parameters on the utility-privacy trade-off of a model for a specific pair of main and private tasks, we need to clearly define the aforementioned costs we will be using throughout this paper. As depicted in Fig 1 and common in practice, a model trained for a biometric task is divided into two parts: First, an embedding network, taking as input an image of a face and returning an *embedding* in a latent space; Second, a head, also called classifier, which is a small network that, given an embedding, returns probabilities that the sample belongs to a given set of classes. The embeddings represent the information extracted from the model before it uses the head to make a prediction on the main biometric task. We study specifically the privacy leakage of the embeddings, the output of the first part of the network, and how much information they leak about a chosen private task.

To measure the utility of such a model, we simply use the **main task accuracy** as our metric. Specifically, for a given biometric task, we measure the proportion of the test set samples that are correctly classified by the model. Ideally, we would like to *maximize* utility, and therefore maximize the accuracy on the main task.

Measuring privacy, however, is more involved, and how to accurately measure the privacy risk of a given system is acknowledged as a one of the core, and arguably most complex, aspects of privacy research. It requires modeling an adequate and realistic adversary by defining what information and resources it has access to, i.e, its capabilities. This is complicated by the difficulty to prove that the chosen adversarial model represents the best possible adversary, and mis-identifying its capabilities could lead to vulnerabilities. Furthermore, even with a good adversary, one needs to define the properties that are measured by the privacy metric, such as uncertainty, information gain or loss, or probability of the adversary's success, among others (Wagner & Eckhoff, 2018).

We study the privacy leakage of the embeddings preceding the classifier portion of the network. Following standard practice, we draw inspiration from the Bayesian-optimal adversary literature (Sablayrolles et al., 2019; Stadler et al., 2024; Chatzikokolakis et al., 2020) to measure privacy. A Bayesian-optimal adversary is the adversary that has the best accuracy on the private task. Ideally, we would like to measure the risk of unintended feature leakage by measuring the probability of success of this optimal adversary $\mathcal{A}(X)$ trying to predict sensitive attribute $Z$ from the set of sensitive classes $\mathbb{Z}$ when given access to embedding $X$ in latent space $\mathbb{X}$. In this formalism, the worst-case risk of unintended feature leakage is thus expressed as

$$\hat{Z}(X) = \underset{\mathcal{A}:\mathbb{X}\to\mathbb{Z}}{\operatorname{argmax}} \mathbb{P}[Z = \mathcal{A}(X)]. \tag{1}$$

This optimal adversary can then be used to measure the privacy risk (or privacy loss) of sensitive attribute $Z$ as

$$PR = \mathbb{P}[Z = \hat{Z}(X)] - \mathbb{P}[Z = \hat{Z}], \tag{2}$$

where $\hat{Z}$ without any argument represents the baseline guess of optimal adversary $\mathcal{A}$ on sensitive attribute $Z$ when it does not have access to any embedding. The privacy risk therefore approaches zero when the optimal adversary's prediction is close to the random guess, which means that observing embedding $X$ reveals virtually no information about sensitive attribute $Z$. Ideally, we would therefore like to *minimize* the privacy risk $PR$, which is equivalent to minimizing $\mathbb{P}[Z = \hat{Z}(X)]$.

In other words, we seek to measure the privacy risk as the **accuracy of an optimal adversary trying to predict the private attribute** given an embedding. In practice, finding the best possible adversary by solving Problem 1 to optimality cannot be guaranteed, and we therefore aim to find the best possible *approximation* of the optimal adversary $\mathcal{A}$. We achieve this by making the assumption that our adversary has query access to the embedding network, i.e., it can query it as many times as desired with face images and obtain the resulting embeddings, without having a white-box view of the model. Additionally, we assume that the adversary has access to a dataset close to the queries at hand, i.e., face images, labeled with various attributes of interest (ethnicity, gender, age, etc.), and uses it in conjunction with the embedding network to learn how to predict private information from face embeddings. Then, given a model trained for a main task, we use hyper-parameter optimization to find the architecture and hyper-parameters of the adversarial model that yield a strong enough adversary. We show empirically in Appendix A.1 that the precise choice of the adversary in this setup has limited impact on our method, as long as it is sufficiently strong.

## 4 EXPERIMENTS

### 4.1 EXPERIMENTAL SET-UP

We use the SMAC3 package (Lindauer et al., 2021) offering the optimization framework described in the previous section. Below, we define the DNN architectures we focus on, as well as the search space and the costs we seek to optimize for, as summarized in Table 1. The SMAC3 configuration we used is detailed in Appendix A.1.1.

Table 1: Experimental Setup and Search Space

| | |
|---:|:---|
| **Experimental Setup** | |
| **Datasets** | FairFace (Kärkkäinen & Joo, 2019), |
| **Embed. Net Architecture** | VGG16 (Simonyan & Zisserman, 2014) |
| **Adversarial Classifier** | FC layers of VGG16, Linear, CosFace (Wang et al., 2018), ArcFace (Deng et al., 2019) |
| **HPO Metrics** | Main Task Accuracy↑ & Private Task Leakage Risk ↓ |
| **# HPO trials** | 80 |
| **Min and Max Budgets (Epochs)** | [5,120] epochs |
| **Drop Ratio** | 3 |
| **Search Space** | |
| **Batch Size** | [30, 100] with a step of size 5 |
| **Head** | {Linear, CosFace (Wang et al., 2018), ArcFace (Deng et al., 2019)} |
| **Loss** | {Cross-Entropy (Mao et al., 2023), Focal Loss (Lin et al., 2018)} |
| **Optimizer** | {Adam (Kingma & Ba, 2017), SGD (Amari, 1993)} |
| **Learning Rate** | $[10^{-5}, 10^{-1}]$ if Adam (log scale) 
 $[10^{-4}, 1]$ if SGD (log scale) |
| **Weight Decay** | $[10^{-6}, 10^{-1}]$ (log scale) |

### 4.1.1 ARCHITECTURE AND DATASET

Following previous work (Li et al., 2019; Singh & Shukla, 2021; Abbasi et al., 2024), we use a CNN architecture, VGG16 (Simonyan & Zisserman, 2014), for the embedding network, with a latent space of size 4608, excluding the final fully-connected layers of the architectures, which we employ as our adversarial classifier and as one of the options for the main task classifier, the other options being CosFace (Wang et al., 2018) and ArcFace (Deng et al., 2019). We additionally include

the following hyper-parameters to our HPO search space: Batch size, loss, optimizer, learning rate, and weight decay.

We evaluate our approach and the baselines on the FairFace dataset (Kärkkäinen & Joo, 2019), which contains 108,501 face images balanced in gender, ethnicity and age. The main and private tasks are therefore chosen from the following soft-biometrics labels: Gender, Ethnicity, and Age, which have 2, 7, and 9 classes, respectively. We run our HPO set-up on 6 different combinations of main and private tasks, using the VGG architecture. In Appendix A.2, we report additional experiments on the Inception-ResNet architecture (Szegedy et al., 2016) and the CelebA dataset (Liu et al., 2015).

### 4.1.2 Adversarial model construction

Before running an HPO for a specific pair of main and private task, we first search for the best possible adversary to measure the privacy risk as accurately as possible, as described in Section 3.2. To do so, we first train the embedding network for its main task for half of the maximum number of epochs, using a random configuration of hyper-parameters, and then use multi-objective HPO to find the a strong enough adversary to measure privacy risk throughout the optimization run.

We acknowledge that this strategy to build an adversary that is used to measure privacy risk does not guarantee finding the best possible adversary. However, it yields a good proxy from which we can draw valuable conclusions. We validate this behavior in Appendix A.1.

We then use these hyperparameters to retrain from scratch the adversarial classifier every time we measure the privacy risk on the private task during the main HPO run. Finally, once we have found the best hyper-parameter configuration for our embedding network through our multi-HPO approach, we re-measure privacy one last time by finding again the best adversary through HPO for our final embedding network.

### 4.2 Baselines

We compare our method against the following baselines:

**Single-Objective HPO**: Our first baseline consists of simply running HPO using the same framework and set-up as for our approach but by optimizing for the **main task accuracy only**, as in traditional HPO. This optimization thus ignores privacy.

**Random Gaussian Noise**: For our second baseline, following common practice, particularly in Federated Learning (Liu et al., 2020; Papernot et al., 2018; Truex et al., 2018), we perturb the embeddings using random Gaussian noise, tuning $\sigma^2$ sufficiently to bring the privacy risk to zero (adversarial accuracy equal to random guess). This gives us a baseline of how much the main task accuracy suffers when we reach an ideal privacy risk under our threat model.

**Differential-Privacy and Siamese Fine-tuning**: For our third baseline, we use the method described in (Osia et al., 2020), which uses a combination of fine-tuning, principal component analysis (PCA) dimensionality reduction and differential-privacy. This method first performs siamese fine-tuning (Chopra et al., 2005) on a model pre-trained for the main task. PCA dimensionality reduction is then applied to the resulting embeddings, before applying noise drawn from a Laplace distribtuion with scale $b = \frac{1}{\epsilon}$ where the privacy budget $\epsilon = 0.5$. We evaluate our method against two versions of this baseline, one with siamese fine-tuning and one without.

**Disentangled Representation Learning**: For our fourth baseline, we employ the Disentangled Representation Learning (DRL) algorithm presented in (Bortolato et al., 2020). This method consists of training a small encoder model (trained in an encoder-decoder fashion) that computes features that disentangle the main task from the private task.

**Adversarial Representation Learning**: Finally, we evaluate our method against the Adversarial Representation Learning (ARL) algorithm presented in (Li et al., 2019). The proposed learning algorithm is designed as a game between an embedding module, a main task classifier and a discriminator (which acts as a proxy adversary), where the embedding module tries to maximize the main task classifier's accuracy while minimizing a discriminator's ability to infer private attributes from the embeddings.

For every baseline, as for our method, we measure privacy by finding the best adversary through HPO for each embedding network.

### 4.3 RESULTS & DISCUSSION

As an example of the typical output of our method, in Fig. 2, we plot the costs of the configurations that were trained during an HPO with the main task of gender recognition and private task of ethnicity, i.e., all configurations that were trained during their successive-halving bracket. The orange points connected by a line represent the Pareto-front of this optimization run: All the points whose costs are Pareto-dominant. A configuration is Pareto-dominant if no objective can be improved without sacrificing at least one other objective. Only configurations that were trained up to the maximal budget are considered as part of the Pareto-Front. Plotting the Pareto-front makes it possible to manually make decisions on what hyper-parameter configuration to chose among the Pareto-dominant ones: The privacy-accuracy trade-off decision is therefore made on a much smaller set. Ideally we would want to reach a point situated in the bottom left of the plot: High main task accuracy and low privacy risk. Note that for the same small range of main task accuracy (between 0.1 and 0.2 on the plot), we have a wide range of privacy risks. This confirms that for the same main task accuracy, there exist models that offer close to no privacy for a given private attribute, as well as models that yield a much more acceptable privacy risk. By simply varying some hyper-parameters during training, we were able to find models with almost the same main task accuracy while offering much more privacy. This suggests that, since different hyper-parameters lead to different local minima, which have been shown to benefit from different properties, such as generalization (Keskar et al., 2017), these local minima may also display different privacy behaviors.

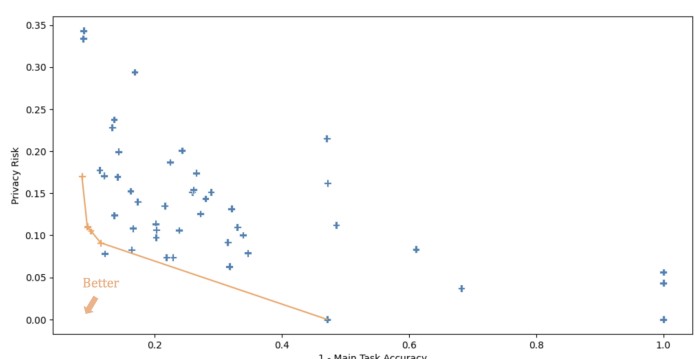

Figure 2: Costs and Pareto-front of the configurations that were trained during one instance of multi-objective hyper-parameter optimization, using the VGG-16 architecture, with gender classification as main task and ethnicity classification as private task.

In Table 2, we detail the main task accuracy and privacy risk, computed as described in Section 3.2, for our method and the baselines, for 6 combinations of main and private tasks described in Section 4.1.

These results first show that, when comparing our multi-objective HPO method to embedding networks that were trained solely to maximize main task accuracy (**HPO**), our approach decreases the privacy risk significantly (by 5 to 20 points) for all but one combinations of main and private tasks. Specifically, our method brings the accuracy of the adversarial classifier significantly closer to random guess, while only suffering from a decrease in main task accuracy of less than 5 points in almost all cases. In other words, our approach maintains a good utility for a much lower privacy risk.

When it comes to the Gaussian noise baseline (**Noisy**), the privacy risk correctly reaches 0, as we tuned the noise variance to achieve this, meaning that the adversary's prediction is equivalent to a random guess on the private class. This, however, comes at a drastic cost in main task accuracy. Indeed, whereas our method yields an average decrease in main task accuracy of 5 points, that of the noise injection baseline reaches 18 points, which would typically preclude the deployment of this strategy in a practical application.

In comparison to the Disentangled Learning baseline (**DL**), our approach yields slightly worse utility but at the gain of better privacy. In fact, the privacy of DL is not much better than that of the HPO baseline, although the latter ignores the private task.

Table 2: Privacy-Utility trade-off comparison with six baselines on the FairFace dataset and VGG16 architecture. The higher the main task accuracy (MA), the better, and the lower the privacy risk (PR), the better. The six baselines are: **HPO**, **Noisy** (Gaussian noise injection), **ARL** (Adversarial representation learning), **DP** (PCA and Differential Privacy), **S-DP** (Siamese Finetuning and Differential Privacy), **DL** (Disentangled Learning). We also evaluate the combination of our method with differential privacy (**MO-HPO+DP**)

| | | HPO | | Noisy | | DL | | ARL | | DP | | S-DP | | MO-HPO (Ours) | | MO-HPO+DP | |
|---|---|---|---|---|---|---|---|---|---|---|---|---|---|---|---|---|---|
| M. Task | P. Task | MA↑ | PR↓ | MA↑ | PR↓ | MA↑ | PR↓ | MA↑ | PR↓ | MA↑ | PR↓ | MA↑ | PR↓ | MA↑ | PR↓ | MA↑ | PR↓ |
| Gender | Ethn. | 0.914 | 0.295 | 0.774 | 0.000 | **0.915** | 0.255 | 0.902 | 0.270 | 0.902 | 0.081 | 0.717 | **0.031** | 0.890 | 0.099 | 0.899 | 0.047 |
| Gender | Age | **0.920** | 0.293 | 0.780 | 0.000 | 0.913 | 0.244 | 0.643 | 0.250 | 0.900 | 0.127 | 0.714 | **0.052** | 0.876 | 0.124 | 0.899 | 0.074 |
| Ethn. | Gender | **0.663** | 0.170 | 0.508 | 0.000 | 0.657 | 0.163 | 0.647 | 0.243 | **0.663** | 0.04 | 0.016 | **0.001** | 0.537 | 0.123 | 0.527 | 0.075 |
| Ethn. | Age | **0.662** | 0.156 | 0.511 | 0.000 | 0.656 | 0.161 | 0.589 | 0.296 | **0.662** | 0.043 | 0.166 | **0.006** | 0.601 | 0.167 | 0.451 | 0.042 |
| Age | Gender | **0.510** | 0.258 | 0.260 | 0.000 | 0.466 | 0.247 | 0.464 | 0.268 | 0.420 | 0.082 | 0.393 | **0.068** | 0.485 | 0.152 | 0.479 | 0.099 |
| Age | Ethn. | **0.508** | 0.213 | 0.259 | 0.000 | 0.471 | 0.208 | 0.372 | 0.381 | 0.427 | 0.057 | 0.394 | 0.051 | 0.489 | 0.114 | 0.330 | **0.033** |

When comparing our method to the Adversarial Representation Learning (**ARL**) algorithm, we observe that not only does the main task accuracy undergoes a larger drop than with our method, but the privacy risk is sometimes even slightly higher than that of models trained with no privacy taken into account. Indeed, although the accuracy of the adversarial classifier on the private task during training decreases significantly, this privacy gain does not transfer to using a freshly trained adversarial classifier to measure privacy. Our method, on the other hand, still maintains the same privacy gain when measuring privacy with freshly trained adversary. Additionally, these results highlight the complexity of choosing appropriate hyper-parameters for ARL algorithms, especially those managing the adversarial loss and the weighs of the different objectives.

We then compare our method to the differential privacy baseline (**DP**), which offers the best privacy-utility trade-off out of our baselines. In most cases, this method provides slightly better privacy than our method (at most five points), as well as a slightly better accuracy, but by still offering a similar privacy-utility trade-off. This is a valuable insight, as it shows that by only tuning hyper-parameters, one can reach similar privacy to a state-of-the-art method using differential privacy. By contrast, the Siamese Fine-tuning variation of DP (**S-DP**) offers very good privacy but at the cost of a significant drop in main task accuracy, making it a suboptimal choice of privacy risk mitigation technique.

Finally, we evaluate a combination of our method and the differential privacy baseline by using the model trained using the hyper-parameter configuration discovered using our Multi-Objective HPO and applying to its embeddings PCA compression, Laplace noise injection (with a smaller scale $b = \frac{1}{\epsilon}$ where the privacy budget $\epsilon = 1.2$), and PCA decompression. This strategy (**MO-HPO-DP**) yields a lower privacy risk than both MO-HPO and DP taken individually, while maintaining utility.

We report main task accuracy and privacy risk for our method and for the baselines on additional datasets and architectures in Appendix A.2.

We additionally investigate whether models trained using different privacy-preserving methods are protected against the inference of unrelated private attributes. For example, does an embedding network trained to maximize its accuracy on gender classification while minimizing the privacy risk with regards to ethnicity also protect against the inference of age? To study this, we use all the embedding networks trained in the previous set of experiments to minimize the privacy risk of a given private attribute and train an adversarial classifier to predict **another unrelated private attribute**. This lets us measure the privacy risk with respect to this unrelated attribute. For our method, the privacy risk for the unrelated task is decreased significantly when compared to a classic embedding network trained with no regards to privacy. For example, a model whose main task is gender classification and trained to protect the private attribute ethnicity also decrease the privacy risk for the private attribute age from **0.440** to **0.248**. This evidences that training a model while trying to decrease the privacy risk with regards to a specific private attribute has a positive impact on the privacy risk of unrelated private attributes. These results suggest that the same hyper-parameters have a consistent impact on the privacy risk. We detail these experiments in Appendix A.3.

### 4.4 EMPIRICAL STUDY OF THE IMPACT OF HYPER-PARAMETERS ON PRIVACY RISK

The many multi-objective HPO runs we have performed across different combinations of main and private tasks have provided us with a large collection of embedding networks and their associated

main task accuracy and privacy risk. We now leverage this data to investigate the impact of hyper-parameters on the risk of unintended feature leakage, which, to our best knowledge, has never been studied before.

To study the impact of hyper-parameters in a more isolated setting, we first consider the models that were trained in a slightly different multi-objective HPO setting. That is, instead of the 6 hyper-parameters described in the experimental setup of Table 1, we reduce the search space to 4 hyper-parameters: Loss, learning rate, optimizer, and batch size. We therefore fix the weight decay parameter to $10^{-3}$ and the head to the FC Linear architecture.

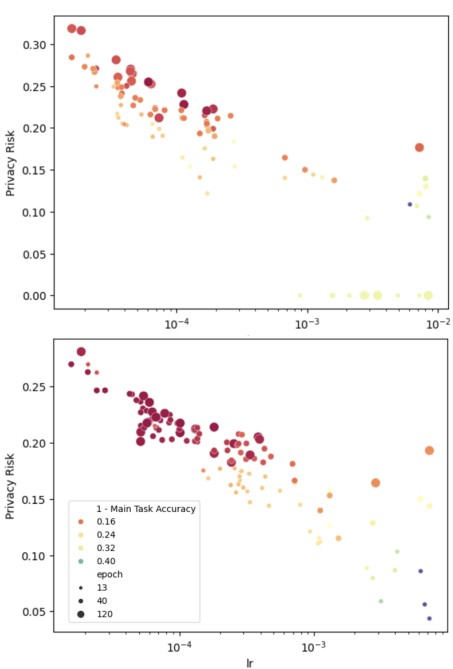

Figure 3: Privacy risk as a function of the learning rate for all models trained using the Adam optimizer during two runs of multi-objective HPO for two different combinations of main and private tasks: Age-Gender (top), and Ethnicity-Age (bottom.) The size of the points indicates for how many epochs the model was trained.

In Fig. 3, we plot the privacy risk as a function of the learning rate for all models trained using the Adam optimizer during two runs of multi-objective HPO for two different combinations of main and private tasks (Age-Gender at the top and Ethnicity-Age at the bottom). The size of the points indicates for how many epochs the model was trained, and the color depicts the performance on the main task (the closer to red, the better). These plots highlight a strong negative correlation between the learning rate and the privacy risk. Indeed, the smaller the learning rate, the higher the privacy risk. We confirm this by computing the Pearson correlation between the learning rate and the privacy risk: -0.685 and -0.647 respectively for both plots. These plots further show that, as expected, the lower the privacy risk, the lower the main task accuracy in general. Nevertheless, some points with lower privacy risk still correspond to excellent main task accuracy; these are the hyper-parameter configurations discovered by our method. The observed trends for these two combinations of main and private task can also be observed across the other combinations we studied. After conducting experiments investigating the impact of weight decay, we come to very similar conclusion, and observe a negative correlation between weight decay and privacy, although less evident. We additionally observe similar the same trends for the other combinations of main and private tasks.

As discussed in Appendix A.4, other hyper-parameters such as batch size, loss, optimizer, seem to have no observable impact on the privacy risk.

## 5 CONCLUSION

In this paper, we have conducted the first multi-objective HPO with the goal of training models that both provide state-of-the-art main task accuracy while defending well against unintended feature leakage. Additionally, we have performed the first study of the impact of hyper-parameters on the risk of unintended feature leakage, and discovered that the learning rate in conjunction with the weight decay have the most impact on privacy, evidencing the good generalization of our method. Overall, we have presented a new approach to training DNNs for biometric tasks in a privacy-preserving way while remaining easier to deploy in practice than previous works. In future we aim to extend our empirical study to larger architectures such as foundation models (Caron et al., 2021; Oquab et al., 2024; Radford et al., 2021) as well as using multi-objective Neural Architecture Search (NAS) instead of HPO, which seeks to automate the search for the best possible architecture for a task at hand (Wang, 2021; Liu et al., 2019; Nekrasov et al., 2019; Yu et al., 2020; 2021), which we suspect will also have significant impact on privacy risk.

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

## A   APPENDIX

### A.1   ADVERSARIAL MODEL CONSTRUCTION: ABLATION STUDY

When constructing the adversarial model to measure privacy throughout the HPO run, as described in 4.1.2, we first train the embedding network for its main task for half of the maximum number of epochs, using a random configuration of hyper-parameters, that we then use to find the best adversary. Fig. 4 shows that training an adversary and measuring the privacy risk on the same embedding network at different training epochs (13, 40 and 120 epochs) yields a similar privacy risk. We observe similar results for other combinations of main and private tasks and architectures. This implies that an embedding network learns information about the private task early in its training, and we can therefore find a sufficiently strong adversary on a embedding network trained for half of the maximum number of epochs. We then freeze this embedding network and run HPO on the adversarial classifier to find the hyper-parameters that maximize the adversary's accuracy on the private task. The search space for the adversary includes learning rate, batch size, loss function and architecture. We limit the architecture of the adversary to the same architecture as the head of the main task, as well as similar architectures (ArcFace, CosFace, Linear). (Li et al., 2019) shows empirically that, in practice, choosing the same architecture for the adversary classifier as the main task classifier (grey-box access) is a good strategy, and leads to a strong adversary. Using this strategy, we are more likely to find hyper-parameters that lead to an adversary that is more accurate on the private task.

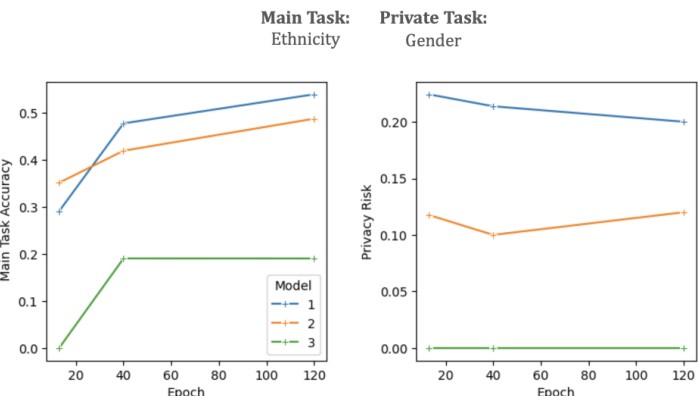

Figure 4: Ablation study of the main task accuracy and privacy risk of 3 embedding networks measured at 3 different training epochs (13, 40, 120) by an adversary trained with the same HP configuration. The embedding network has the VGG-16 architecture and is trained with the main task

We argue that our method of constructing an adversarial model yields a good proxy from which we can draw valuable conclusions in Fig.5. It shows that for 5 models with the same architecture trained on the same main task but with different hyper-parameters, different adversaries with different architectures and trained with different hyper-parameters (1) always output, for the same embedding network, a similar privacy risk with a small standard deviation, (2) will always rank the embedding networks by privacy risk in a very similar manner, preserving the relative ordering of the models. These two properties are crucial for our multi-HPO approach, as we do not need guarantees of having the best possible adversary to chose the models that offer a better privacy than the other models trained.

### A.1.1   SMAC3 CONFIGURATION

For each run, we aim to maximize the accuracy on the main task while minimizing the adversary's accuracy on the private task. We run HPO with a maximum budget of 120 epochs and for 80 trials (a trial being the training from scratch *or* the intensification of a model for which the training has already been done for a limited budget). Finally we choose a drop ratio of 3, meaning that, every

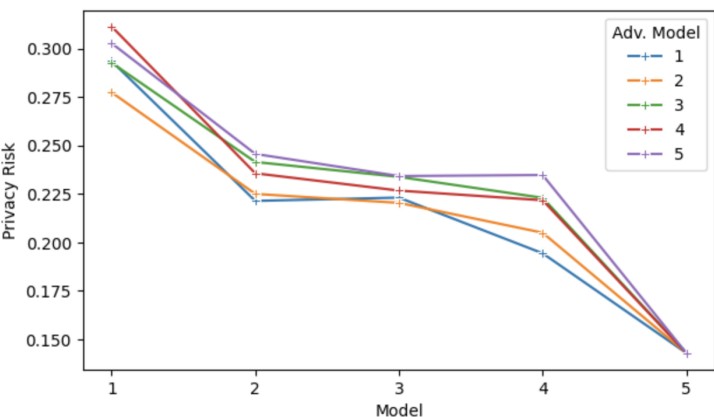

Figure 5: Ablation study of the privacy risk of 5 embeddings networks as measured by 5 adversarial models trained with randomly sampled HP configurations.

1/3 of the maximum number of epochs, we get rid of 1/3 of the remaining configurations in the bracket. The optimizer finally outputs a list of the Pareto-dominant configurations in terms of main task accuracy and privacy risk.

## A.2 ADDITIONAL EXPERIMENTS: VISUAL EXAMPLES AND ADDITIONAL DATASETS AND ARCHITECTURES

We first showcase visual examples of how our method compares to two baselines in Fig. 6. The input images are passed to three embedding networks; one trained using our multi-objective HPO method, one using HPO to maximize the main task accuracy only, and the last one implementing the gaussian noise injection baseline. The embedding obtained with our approach still yields a correct main task prediction, even though the embedding network was trained in a privacy-preserving manner (the genders of the input images are still predicted accurately). However, this embedding successfully fools a strong adversarial classifier, as opposed to that obtained with the baseline, which reveals the ethnicity of the input image, thus resulting in a higher privacy risk. On the other hand, the noise injection baseline also provides a very low privacy risk but suffers from a significant loss in utility (the genders of the input images are no longer successfully predicted in most cases). We can observe a similar privacy-utility trade-off for these three methods with another combination of private and main tasks, age and gender, depicted on the right of Fig. 6.

| | | | | | | | |
|---|---|---|---|---|---|---|---|
| Main Task Label | Female | Male | Male | Main Task Label | 30-39 | 3-9 | 40-49 |
| Priv. Task Label | White | East-Asian | Black | Priv. Task Label | Female | Female | Female |
| **HPO - Good Utility  Poor Privacy** | | | | **HPO - Good Utility  Poor Privacy** | | | |
| Main Task Pred. | Female ✓ | Male ✓ | Male ✓ | Main Task Pred. | 30-39 ✓ | 3-9 ✓ | 40-49 ✓ |
| Priv. Task Pred. | White ✓ | East-Asian ✓ | Black ✓ | Priv. Task Pred. | Female ✓ | Female ✓ | Female ✓ |
| **Noisy - Poor Utility  Good Privacy** | | | | **Noisy - Poor Utility  Good Privacy** | | | |
| Main Task Pred. | Male ✗ | Female ✗ | Male ✓ | Main Task Pred. | 3-9 ✗ | 20-29 ✗ | 40-49 ✓ |
| Priv. Task Pred. | Black ✗ | M. Eastern ✗ | Latino ✗ | Priv. Task Pred. | Male ✗ | Male ✗ | Male ✗ |
| **Multi-Obj HPO (Ours)  Good Utility  Good Privacy** | | | | **Multi-Obj HPO (Ours)  Good Utility  Good Privacy** | | | |
| Main Task Pred. | Female ✓ | Male ✓ | Male ✓ | Main Task Pred. | 30-39 ✓ | 3-9 ✓ | 40-49 ✓ |
| Priv. Task Pred. | Black ✗ | M. Eastern ✗ | Latino ✗ | Priv. Task Pred. | Male ✗ | Male ✗ | Male ✗ |

Figure 6: Example of the main and private task inference from models trained using classic HPO, the noise injection baseline, and our multi-objective HPO method.

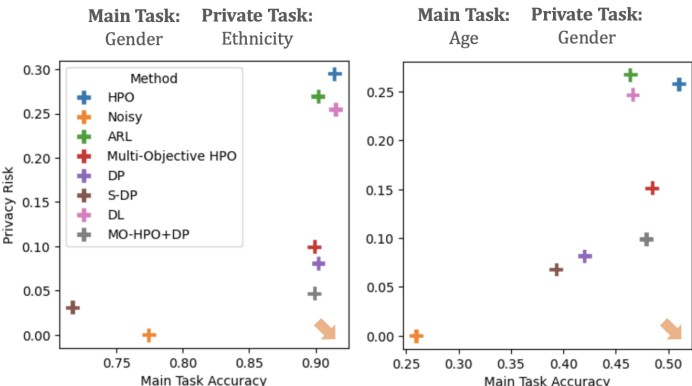

Figure 7: Privacy-utility trade-off of our method (Multi-Objective HPO ✚) and the combination of our method and the differential privacy mechanism (MO-HPO+DP ✚) and the baselines for two combinations of main and private task on thr VGG-16 Architecture. The arrow represents the best possible privacy-utility trade-off: High main task accuracy and low privacy risk.

We then illustrate the privacy-utility trade-off of our method compared to the baselines in Fig. 7, for two combinations of main and private tasks. Ideally, we would like to reach both high main task accuracy and low privacy risk, i.e., be as close as possible to the lower right corner of the plot, as indicated by the arrow. The models trained using our multi-HPO method, depicted in red, offer a very good privacy-utility trade-off, in a similar range as the DP baseline, depicted in purple, whereas the combination of our method and DP clearly yields the best privacy-utility trade-off.

We then report results of experiments conducted in the same manner and against the same 6 baselines, but using additional architectures and datasets. We first evaluate our method using the same dataset, FairFace but using a different architecture for the embedding network: Inception-ResNet (Szegedy et al., 2016), with a latent space of size 512. We report the results for 6 combinations of main and private tasks in Table 3.

Table 3: Privacy-Utility trade-off comparison with four baselines on the FairFace dataset and Inception Res-Net architecture. The higher the main task accuracy (MA), the better, and the lower the privacy risk (PR), the better.

| M. Task | P. Task | HPO MA ↑ | HPO PR ↓ | Noisy MA ↑ | Noisy PR ↓ | DL MA ↑ | DL PR ↓ | DP MA ↑ | DP PR ↓ | MO-HPO (Ours) MA ↑ | MO-HPO (Ours) PR ↓ | MO-HPO+DP MA ↑ | MO-HPO+DP PR ↓ |
|---|---|---|---|---|---|---|---|---|---|---|---|---|---|
| Gender | Ethnicity | 0.915 | 0.036 | 0.814 | 0.000 | **0.918** | 0.09 | 0.888 | 0.002 | 0.908 | 0.029 | 0.912 | 0.025 |
| Gender | Age | 0.914 | 0.112 | 0.809 | 0.001 | **0.918** | 0.143 | 0.887 | 0.061 | 0.887 | 0.100 | 0.877 | 0.090 |
| Ethnicity | Gender | **0.651** | 0.080 | 0.371 | 0.000 | 0.646 | 0.161 | 0.535 | 0.026 | 0.623 | 0.083 | 0.588 | 0.061 |
| Ethnicity | Age | **0.649** | 0.085 | 0.368 | 0.000 | 0.500 | 0.0713 | 0.535 | 0.014 | 0.613 | 0.014 | 0.636 | 0.007 |
| Age | Gender | **0.549** | 0.216 | 0.219 | 0.036 | 0.509 | 0.236 | 0.348 | 0.068 | 0.513 | 0.150 | 0.517 | 0.109 |
| Age | Ethnicity | 0.545 | 0.144 | 0.220 | 0.021 | **0.547** | 0.166 | 0.343 | 0.027 | 0.527 | 0.103 | 0.518 | 0.049 |

We then evaluate our method on the VGG16 architecture but this time using the CelebA dataset (Liu et al., 2015), which contains 202,599 face images with 40 labeled soft-biometrics. Because of the varying label quality of the CelebA dataset, we select 3 categories that have been shown to have high consistency across annotators (Wu et al., 2023): Gender, Gray Hair, Glasses, wich we use as options for main and private tasks. All three categories are binary classes. We report the results for 4 combinations of main and private tasks in Table 4.

## A.3 ADDITIONAL EXPERIMENTS: TRANSFERABILITY

In this section, we investigate whether models trained using different privacy-preserving methods are protected against the inference of unrelated private attributes. For example, does an embedding network trained to maximize its accuracy on gender classification while minimizing the privacy risk with regards to ethnicity also protect against the inference of age? To study this, we use all the em-

Table 4: Privacy-Utility trade-off comparison with four baselines on the CelebA dataset on the VGG16 architecture. The higher the main task accuracy (MA), the better, and the lower the privacy risk (PR), the better.

| | | HPO | | Noisy | | DL | | DP | | MO-HPO (Ours) | | MO-HPO+DP | |
|---|---|---|---|---|---|---|---|---|---|---|---|---|---|
| M. Task | P. Task | MA ↑ | PR ↓ | MA ↑ | PR ↓ | MA ↑ | PR ↓ | MA ↑ | PR ↓ | MA ↑ | PR ↓ | MA ↑ | PR ↓ |
| Gender | Eyeglasses | **0.989** | 0.367 | 0.822 | 0.000 | 0.985 | **0.218** | 0.980 | 0.223 | 0.986 | 0.228 | 0.985 | **0.218** |
| Gender | Gray Hair | **0.989** | 0.353 | 0.825 | 0.000 | 0.986 | 0.189 | 0.980 | 0.189 | 0.986 | **0.187** | 0.985 | 0.189 |
| Eyeglasses | Gender | **0.996** | 0.225 | 0.801 | 0.000 | 0.992 | 0.188 | 0.935 | **0.064** | 0.991 | 0.224 | 0.991 | 0.188 |
| Gray Hair | Gender | **0.959** | 0.393 | 0.500 | 0.180 | 0.951 | 0.002 | 0.855 | 0.242 | 0.951 | **0.001** | 0.951 | **0.001** |

bedding networks trained in the previous set of experiments to minimize the privacy risk of a given private attribute and train an adversarial classifier to predict **another unrelated private attribute**. This lets us measure the privacy risk with respect to this unrelated attribute. We expose our findings in Table 5. For both our method and the ARL baseline, the privacy risk for the unrelated task is decreased significantly when compared to a classic embedding network trained with no regards to privacy. This evidences that training a model while trying to decrease the privacy risk with regards to a specific private attribute has a positive impact on the privacy risk of unrelated private attributes. Surprisingly, in most cases, the ARL baseline offers more privacy guarantees with respect to the unrelated private attribute than the private attribute whose inference it was actually trained to defend against. The transferability of the selected hyper-parameter configurations to other private tasks also suggests that the same hyper-parameters have a consistent impact on the privacy risk.

Table 5: Leakage of a private attribute unrelated to the private attribute used to train the model. The lower the privacy risk, the better.

| | | | HPO | ARL | **Multi-Obj. HPO (Ours)** |
|---|---|---|---|---|---|
| Main Task | Private Task | Unrelated Priv. Task | Priv. Risk ↓ | Priv. Risk ↓ | Priv. Risk ↓ |
| Gender | Ethnicity | Age | 0.440 | 0.271 | **0.248** |
| Gender | Age | Ethnicity | 0.438 | **0.177** | 0.218 |
| Ethnicity | Gender | Age | 0.267 | 0.260 | **0.242** |
| Ethnicity | Age | Gender | 0.670 | **0.274** | 0.680 |
| Age | Gender | Ethnicity | 0.356 | **0.245** | 0.276 |
| Age | Ethnicity | Gender | 0.758 | **0.242** | 0.679 |

## A.4 ADDITIONAL EXPERIMENTS: EMPIRICAL STUDY

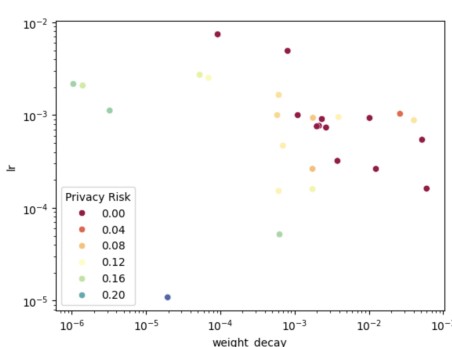

Figure 8: Privacy risk as a function of both learning rate and weight decay for all models trained during a run of multi-objective HPO on the VGG-16 model for the combination of main and private tasks Gender-Ethnicity. The color indicates the privacy risk (the closer to red, the better).

In Fig. 9a, we show the effect of bacth size, loss and optimizer on privacy risk for the models trained during a multi-objective HPO run with age as the main task and gender as the private task. These plots do not show any correlation between these three hyper-parameters and the privacy risk, which is confirmed by very small Pearson correlations (-0.056, -0.039, and 0.204) between the privacy risk and each of the three hyper-parameters. The same trends are also observable for other main and private task combinations.

We now investigate the impact of the two hyper-parameters that were left out in the previous experiments, weight decay and head architecture, on the privacy risk and how they interact with the other hyper-parameters we already studied. That is, we go back to our initial search space of 6 hyper-parameters described in Table 1, and use the models from the 6 multi-objective HPO runs we conducted in Section 4.3. In Fig. 9b, we plot the privacy risk

as a function of the head, weight decay, and learning rate for gender classification as the main task and ethnicity as the private attribute. The first plot in the figure evidences that the choice of the head has some impact on both privacy and main task accuracy. Indeed ArcFace seems to provide lower privacy risk than the other two head architectures, but at the cost of a lower main task accuracy.

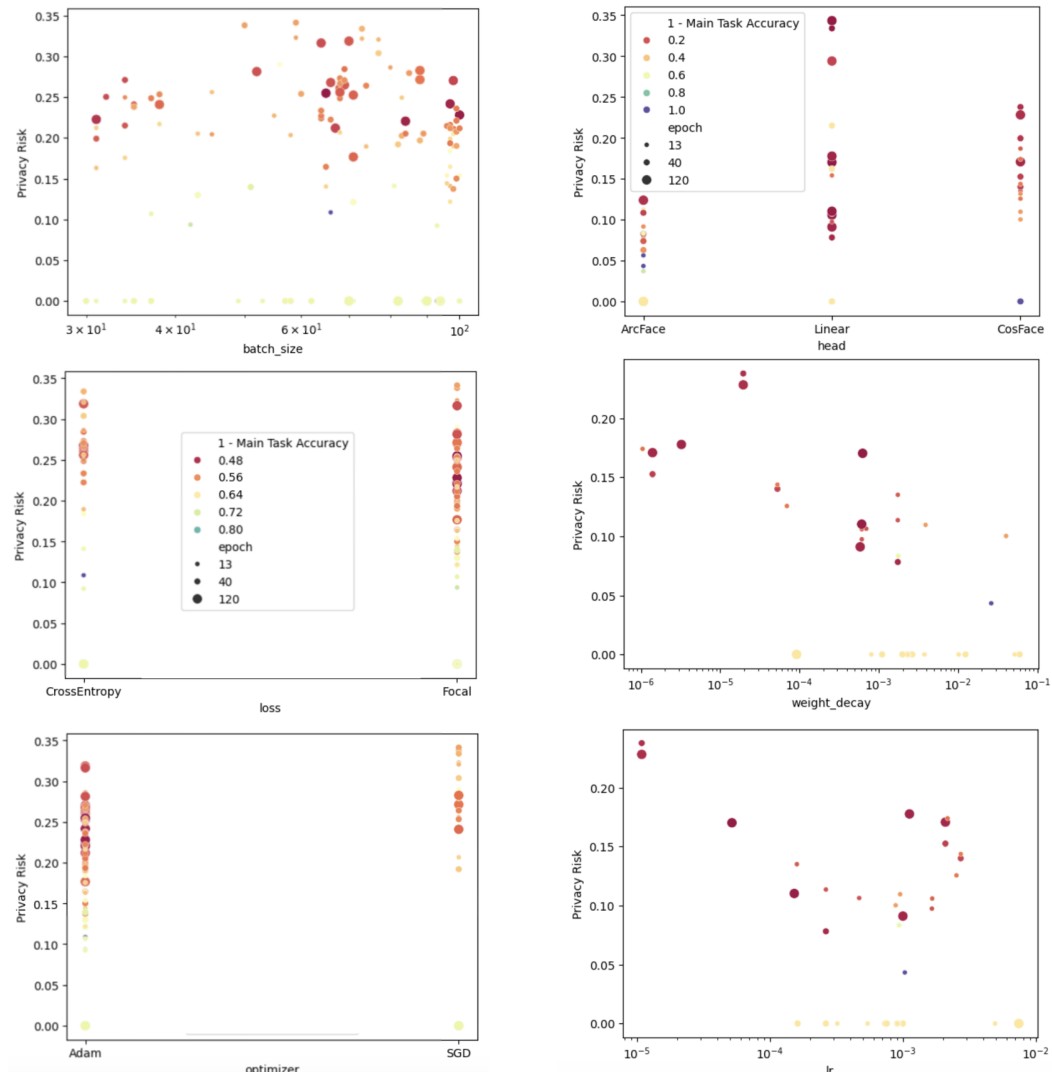

(a) Privacy risk as a function of the batch size, loss, and optimizer for all models trained during a run of multi-objective HPO on the VGG-16 model for the combination of main and private tasks Age-Gender. The size of the points indicates how many epochs the model was trained, and the color shows performance on the main task (the closer to red, the better).

(b) Privacy risk as a function of the head, weight decay, and learning rate for all models trained during a run of multi-objective HPO on the VGG-16 model for the combination of main and private tasks Age-Gender. Point size indicates training epochs, color shows main task performance.

Figure 9: Privacy risk under different hyperparameter combinations during multi-objective HPO on VGG-16 for the Age-Gender task combination.

The Linear and CosFace head architectures provide similar main task accuracy, but the Linear head yields a larger range of privacy risks, ultimately resulting in models with a lower privacy risk than CosFace for a higher main task accuracy. The second plot shows a similar trend as for the learning rate, with a negative correlation between weight decay and privacy risk, i.e., a larger weight decay

leads to a lower privacy risk. On the third plot, we can observe again that the learning rate has an impact on the privacy risk, although the trend is less evident than in Fig. 3, which showed a clear relationship between learning rate and privacy risk. This can be explained by the fact that the models depicted here vary in terms of both learning rate and weight decay, which makes it harder to study the impact of each individual hyper-parameter on the privacy risk. Indeed, it is in fact the combination of both that impacts the privacy risk the most, as depicted in Fig. 8, which shows the privacy risk (represented by color) as a function of both the learning rate and privacy risk. This plot shows that a combination of a larger weight decay and learning rate leads to better privacy. Additionally, for a fixed weight decay of $10^{-3}$, varying the learning rate is clearly very negatively correlated with the privacy risk, which is exactly what we had observed in Fig. 3. We have also observed the same trends for the other combinations of main and private tasks.

