# OpenReview forum: "On the Impact of Hyper-Parameters on the Privacy of Deep Neural Networks"
_ICLR.cc/2026/Conference — ICLR 2026 Conference Withdrawn Submission_

### Official Review · Reviewer_naFt · 2025-10-16

**Soundness:** 2
**Presentation:** 1
**Contribution:** 1
**Rating:** 2
**Confidence:** 4

**Summary:**

This paper investigates the problem of "unintentional feature leakage" in deep neural networks, where embeddings produced for a primary task inadvertently reveal sensitive, private attributes. The authors propose that training hyper-parameters, which are often tuned solely for accuracy, have an impact on this privacy leakage.

**Strengths:**

- The paper addresses a critical issue in ML privacy.

- The authors provide a Pareto front of models that offer an optimal trade-off between utility and privacy.

**Weaknesses:**

- It offers limited insight into why relationships between hyper-parameters (learning rate, weight decay) and privacy risk exist. A deep discussion on this one would elevate the paper's contribution.

- The proposed approach consists of applying a standard MO-HPO framework (SMAC3) to a new problem domain. The paper does not introduce any new modifications to the optimization process, nor does it address whether there are any challenges when applying the framework to solve privacy risks.

- The paper mentions its problem as "unintentional feature leakage" but fails to connect it to the well-established research field of Attribute Inference Attacks. This is a significant omission. There is a line of work in the field of Attribute Inference Attacks already discussing such privacy risks, and these papers are not included in the related work section.

- The paper's evaluation of privacy risk is limited to a single, fixed adversary model. The paper does not explore how the proposed method would be against various unintentional feature leakage attacks or various Attribute Inference Attacks.

- The structure in Section 4.4 makes it difficult for the reader to follow the paper's main conclusion. The presentation should be clear on which hyper-parameters cause privacy risk.

**Questions:**

See coments above.

---

### Official Review · Reviewer_7wPS · 2025-10-30

**Soundness:** 3
**Presentation:** 3
**Contribution:** 3
**Rating:** 6
**Confidence:** 3

**Summary:**

This paper presents a comprehensive investigation into the relationship between standard training hyper-parameters and the unintended leakage of private attributes. The authors propose a multi-objective, multi-fidelity Hyper-Parameter Optimization (HPO) framework to find models that balance main task utility with privacy risk. A key finding is that higher learning rates and weight decay consistently lead to lower privacy leakage, a valuable empirical insight. The paper is methodologically sound, features extensive experiments, and makes a strong case for "privacy-by-design" through a highly generalizable, architecture-agnostic approach.

**Strengths:**

1. The paper tackles a unexplored problem about how hyper-parameters affect the privacy of the resulting model, and It is a more direct privacy concern for deployed systems.
2. The use of multi-objective HPO is a powerful and elegant choice. The utility-privacy Pareto frontier can often find configurations with significantly better privacy at a minimal cost to utility.
3. The paper benchmarks against a wide array of strong and diverse baselines, including adversarial training (ARL), disentangled learning (DL), and differential privacy (DP).
4. Experiments are conducted across multiple datasets (FairFace, CelebA), architectures (VGG16, Inception-ResNet), and task combinations, convincingly demonstrating the generality of the findings.
5. The inclusion of a "transferability" experiment (Section 4.3 and Appendix A.3) is a particularly insightful addition, showing that hyper-parameters chosen to protect one private attribute also improve privacy for other, unrelated attributes.
6. Higher learning rates and weight decay consistently reduce privacy risk is a major contribution. The analysis in Section 4.4 and the appendices effectively visualizes and validates this relationship.

**Weaknesses:**

1. Beyond the empirical correlation, what is your hypothesis for the underlying reason that higher learning rates and weight decay lead to more privacy-preserving models? Do you think it's related to the flatness of the minima or another property of the optimization trajectory?
2. Can you elaborate on the intuition behind this combination? Does the MO-HPO step create a representation that is inherently more robust to the noise introduced by the subsequent DP mechanism?
3. Would the method generalize to other domains with sensitive attributes, such as medical imaging or financial data? What factors might affect this generalizability?
4. In the transferability experiments, your method improves privacy for unrelated attributes, but the ARL baseline sometimes does even better. It needs further exploring and discussion.

**Questions:**

1. Beyond the empirical correlation, what is your hypothesis for the underlying reason that higher learning rates and weight decay lead to more privacy-preserving models? Do you think it's related to the flatness of the minima or another property of the optimization trajectory?
2. Can you elaborate on the intuition behind this combination? Does the MO-HPO step create a representation that is inherently more robust to the noise introduced by the subsequent DP mechanism?
3. Would the method generalize to other domains with sensitive attributes, such as medical imaging or financial data? What factors might affect this generalizability?
4. In the transferability experiments, your method improves privacy for unrelated attributes, but the ARL baseline sometimes does even better. It needs further exploring and discussion.

**Details Of Ethics Concerns:**

No Ethics Concerns

---

### Official Review · Reviewer_f1nh · 2025-11-01

**Soundness:** 3
**Presentation:** 1
**Contribution:** 2
**Rating:** 2
**Confidence:** 3

**Summary:**

This paper studies how training hyper-parameters affect unintended feature leakage from deep networks used as embedding/feature extractors for biometric tasks (e.g., gender/ethnicity/age). The authors (i) formulate privacy risk as the (approximate) accuracy of a strong adversary that maps embeddings → private attribute, (ii) run the first multi-objective, multi-fidelity HPO campaign that jointly optimizes main-task accuracy and this empirical privacy risk (using SMAC3 / HyperBand + ParEGO), and (iii) use the resulting dataset of trained models to (a) show that substantial reductions in privacy leakage can be found with negligible utility loss, (b) identify which hyper-parameters (notably learning rate and weight decay) most strongly correlate with privacy risk, and (c) demonstrate some transfer of privacy-preserving hyper-parameter settings across different main/private task pairs. Experiments are primarily on FairFace (VGG16; additional results on Inception-ResNet and CelebA in appendix). The authors compare MO-HPO to baselines including single-objective HPO, Gaussian-noise, DP+PCA, disentangled / adversarial representation methods, and a hybrid MO-HPO+DP

**Strengths:**

1) Practical, deployment-oriented framing. The paper addresses a real-world, operationally important question: can one reduce leakage of sensitive attributes from embeddings without changing architecture or retraining complex adversarial setups, simply by choosing different hyper-parameters?

2) Comparative evaluation vs many baselines.

3) The authors adopt a Bayesian-optimal-adversary-inspired metric (adversary accuracy over random baseline) and take care to tune an adversarial classifier via HPO so the measured privacy risk is meaningful rather than an artifact of a weak probe. This is stronger than simply training a single probe network.

**Weaknesses:**

1) Just analyzing what hyperparameters are best is sort of not enough for ICLR paper, imho. The paper should come up with novelty, not just  technically reporting results of different experiments.

2) Compute cost and practicality of MO-HPO. Multi-objective HPO with an inner loop that retrains adversaries can be expensive. Report total GPU-hours per MO-HPO run, and compare wall-clock / computational cost against baseline defenses (e.g., DP pipeline or ARL) so practitioners can weigh benefits vs cost.

3) The paper focuses on training hyper-parameters (batch size, LR, optimizer, loss, weight decay) but keeps architecture fixed (VGG16 mainly). While the authors motivate this choice (deployment ease), architecture and capacity are known to influence leakage (and fairness). Discuss the likely effect of including architecture (or report a small experiment where architecture is varied) and whether “privacy-preserving hyper-parameters” generalize across backbones.

4) Lots of space could be saved in main text without huge white spaces in Figures 1,2,3 and Table 1.

**Questions:**

How much do privacy-risk estimates change if (a) you increase/decrease the adversary’s HPO budget?

---

### Note · Authors · 2025-11-24

I have read and agree with the venue's withdrawal policy on behalf of myself and my co-authors.